# Computer-Aided Diagnosis of COVID-19 from Chest X-ray Images Using Hybrid-Features and Random Forest Classifier

**DOI:** 10.3390/healthcare11060837

**Published:** 2023-03-13

**Authors:** Kashif Shaheed, Piotr Szczuko, Qaisar Abbas, Ayyaz Hussain, Mubarak Albathan

**Affiliations:** 1Department of Multimedia Systems, Faculty of Electronics, Telecommunication and Informatics, Gdansk University of Technology, 80-233 Gdansk, Poland; 2College of Computer and Information Sciences, Imam Mohammad Ibn Saud Islamic University (IMSIU), Riyadh 11432, Saudi Arabia; qaabbas@imamu.edu.sa; 3Department of Computer Science, Quaid-i-Azam University, Islamabad 44000, Pakistan

**Keywords:** computer-aided detection system, smart healthcare system, deep learning, COVID-19, random forest, statistical features, convolutional vision transformers

## Abstract

In recent years, a lot of attention has been paid to using radiology imaging to automatically find COVID-19. (1) Background: There are now a number of computer-aided diagnostic schemes that help radiologists and doctors perform diagnostic COVID-19 tests quickly, accurately, and consistently. (2) Methods: Using chest X-ray images, this study proposed a cutting-edge scheme for the automatic recognition of COVID-19 and pneumonia. First, a pre-processing method based on a Gaussian filter and logarithmic operator is applied to input chest X-ray (CXR) images to improve the poor-quality images by enhancing the contrast, reducing the noise, and smoothing the image. Second, robust features are extracted from each enhanced chest X-ray image using a Convolutional Neural Network (CNNs) transformer and an optimal collection of grey-level co-occurrence matrices (GLCM) that contain features such as contrast, correlation, entropy, and energy. Finally, based on extracted features from input images, a random forest machine learning classifier is used to classify images into three classes, such as COVID-19, pneumonia, or normal. The predicted output from the model is combined with Gradient-weighted Class Activation Mapping (Grad-CAM) visualisation for diagnosis. (3) Results: Our work is evaluated using public datasets with three different train–test splits (70–30%, 80–20%, and 90–10%) and achieved an average accuracy, F1 score, recall, and precision of 97%, 96%, 96%, and 96%, respectively. A comparative study shows that our proposed method outperforms existing and similar work. The proposed approach can be utilised to screen COVID-19-infected patients effectively. (4) Conclusions: A comparative study with the existing methods is also performed. For performance evaluation, metrics such as accuracy, sensitivity, and F1-measure are calculated. The performance of the proposed method is better than that of the existing methodologies, and it can thus be used for the effective diagnosis of the disease.

## 1. Introduction

The coronavirus of 2019 is a severe illness that spreads quickly. A recently identified virus known as the severe acute respiratory syndrome coronavirus-2 (SARS-CoV-2) is the cause of it. This disease first appeared in December 2019 in Wuhan, Hubei Province, China. Since then, it has spread quickly to almost every country and region in the world, causing a global pandemic that has never happened before [1]. The Chinese outbreak of COVID-19 was declared a public health emergency of international concern (PHEIC) by the Director of the World Health Organization (WHO) on 30 January 2020, as it posed a serious risk to states with vulnerable and fragile health organisations; the outbreak was subsequently recognised as a pandemic by the WHO in March 2020 [2]. Recently, on 26 November 2021, a new variant of omicron was diagnosed in people in South Africa, which is still under epidemiological study to understand the impact of omicron. Around 4.8 million new cases and 39,000 deaths were reported globally in the last 28 days (30 January to 26 February 2023), representing a 76% and 66% decline, respectively, from the previous 28 days. Around 758 million confirmed cases and over 6.8 million deaths had been reported globally as of 26 February 2023 [2].

According to the WHO, the new variant B.1.1.529, called omicron, is still concerning [3]. Researchers in the UK, South Africa, Denmark, and all over the globe have investigated many elements of omicron. The outcomes of these studies will be shared when they become available. There is no evidence that the symptoms associated with omicron are distinct from those associated with other variants. All COVID-19 variants, including the worldwide predominant delta variant, can cause severe illness or even death, especially in the most vulnerable; thus, prevention is the best approach. As of 22 December 2021, the omicron variant was present in 110 countries across all six WHO regions. Our understanding of the omicron variety continues to evolve as further information becomes available. Based on the available research, omicron has a substantial growth advantage over delta. In locations where community transmission has been documented, the virus spreads significantly faster than the delta strain, with a doubling period of only two to three days. Estimated growth rates in South Africa are now falling, owing primarily to falling rates in the Gauteng province. It is unclear if the reported high growth rate since November 2021 is due to immune evasion or intrinsic increased transmissibility, although it is most likely a combination of the two. Early evidence from South Africa, the United Kingdom (UK), and Denmark suggest that omicron has a lower risk of hospitalisation than delta. However, hospitalisation risk is only one component of severity that admission policies can influence. More data from different jurisdictions is needed to determine how clinical severity criteria, such as oxygen consumption, mechanical ventilation, and death, are linked to omicron. In the fall of 2022, the omicron subvariant of COVID-19, BA.5, became one of the virus’s dominant strains in the United States. It was the most easily transmitted strain at the time, and it is capable of evading immunity from COVID infection and immunisation [4,5,6].

Vaccines are currently available but are still undergoing testing against the omicron variant. In the United Kingdom, after two doses of either the Pfizer BioNTech-Comirnaty or AstraZeneca-Vaxzervria vaccines, the effectiveness of the omicron vaccination against symptomatic disease decreased significantly compared to that of the delta vaccine. However, two weeks after obtaining a Pfizer BioNTech-Comirnaty booster, the effectiveness against delta was marginally increased or comparable. More research is necessary to assess the effect of booster vaccination on the durability of protection against severe and mild disease, infection, and transmission, particularly in the context of evolving variants [7]. Data on the effects of this novel variant of concern on vaccine effectiveness, particularly against severe disease, are currently lacking [8,9]. As a result, visual clues can be utilised as an alternate method for rapidly screening infected patients. The most prevalent symptom of this virus is a lung infection, for which CXR images are widely used as a visual indicator [10]. Initial assessment, detection, and cure of patients with suspected or proven COVID-19 diseases could be possible with radiological imaging modalities such as digital chest X-ray images and thoracic computed tomography (CT).

Although traditional diagnosis has gotten more precise over time, it is still vulnerable to medical personnel risks. It is also more expensive as diagnostic test kits are required for each patient. In comparison, medical imaging procedures such as CT scans and X-rays, which are considered safer, faster, and more easily accessible, can be employed for screening. For COVID-19 screening, X-ray image screening is preferred over CT scans since it is more widely available, easy, fast, and substantially less expensive [11,12]. On the other hand, manual identification of COVID-19 using CXR images might be time-consuming. There may be several inaccuracies and human flaws if there is no prior experience with or knowledge of the virus and its symptoms. To tackle the COVID-19 worldwide pandemic, the US federal government pushed health specialists and academics to adopt machine learning (ML), artificial intelligence (AI), and other developing applications in March 2020 [13]. As a result, there is a need to automate such operations across the board. It should be available to make diagnosis more effective, precise, and rapid. The existing work on COVID-19 detection using machine learning and deep learning (DL) is discussed in Section 2 of this paper. A visual example of chest X-ray images is shown in Figure 1.

### 1.1. Current Limitations and Major Contributions

Several DL and pre-trained DL model were used to detect COVID-19 using chest X-ray images. However, these methods focused only on performance improvement. Most of the models constructed are limited by the amount of COVID-19 images, especially at the onset of the pandemic, because of unavailability or limited access to publicly available data [14]. Moreover, most of the developed methods include several deep learning models, making these methods computationally complex. In addition, many of these developed methods focus on enhancing the quality of CXR images before applying them to the DL-based model. Therefore, there is still a gap in obtaining precise chest X-ray images for COVID-19 detection using a machine learning algorithm. Our proposed comprehensive research work focuses on diagnosing COVID-19 using CXR images. The main purpose of this study is to develop a simple approach for COVID-19 detection based on ML algorithms. Our work comprises pre-processing, feature extraction, and classification phases to detect normal X-rays and COVID-19 images.

Following are the contributions of our proposed work.

We have developed a new feature framework based on a convolutional vision transformer and an optimal set of GLCM features, such as contrast, energy, entropy, and correlation, that is computationally efficient for extracting compelling features from enhanced chest X-ray images.A cost-efficient and simple pre-processing method was implemented based on the Gaussian filter and logarithmic operator.The classification of conventional X-ray and COVID-19 images was accomplished via a random forest classifier with three distinct training–test split techniques.Results were evaluated on these performance metrics: accuracy, precision, recall, and F1 score. A comparative assessment of the proposed work with existing similar work for COVID-19 detection is also presented.

### 1.2. Paper Organization

This paper is structured as follows: Section 2 summarises the current work on COVID-19 detection. Section 3 describes the data acquisition and the proposed method for the diagnostic system. Section 4 presents the experimental study of the proposed work. Section 5 presents the discussion, and, finally, Section 6 concludes the paper with future remarks.

## 2. Literature Review

After the COVID-19 outbreak in late 2019, more and more scientists, researchers in medical image analysis, and AI experts have shown interest in making a very good COVID-19 diagnostic system that uses chest X-rays. There are several studies based on AI that the researcher developed to help clinicians efficiently detect COVID-19 pneumonia [14,15,16,17,18]. Several research and development-based methods have addressed chest X-ray image classification using DL techniques to aid in the detection of pneumonia and COVID-19 disorders. Ref. [19] presents a good review study about DL-based COVID-19 detection method using CXR images. Another [20] study examines existing deep learning methods for detecting coronavirus infection in lung images. The study in [12] proposed using chest X-ray images to investigate COVID-19 using a patch-based deep learning method. They used a fully connected DenseNet, composed of 103 convolutional layers, to segment the lung region from CXR images. After that, a few random patches were taken from the lung areas that had been split up and put into the multimodal machine learning classifier as inputs. The chest X-rays used in this study came from healthy people and people with bacterial pneumonia, tuberculosis, and coronavirus-related viral pneumonia This proposed work achieved an overall detection accuracy of 88.9% and an F1-score of 84.4%.

Another work [21] proposed a method that utilises a deep learning architecture called DarkCovidNet(DCN). They used digital chest X-rays images to automatically identify COVID-19. Their model had 17 convolutional layers and could handle both binary (non-COVID vs. COVID-19) and multi-classification (non-COVID-19 viral pneumonia vs. COVID-19 vs. bacterial pneumonia). For binary classifications, the diagnostic accuracy was found to be 98.08%, and for multinomial classifications, it was found to be 87.02%. In a similar study [22], a DL model (Inf-Net) was used to segment the suspicious area of the lung using CXR samples to look for COVID. The suspicious area was chosen at random. In this work, the final segmented maps are made with the help of a parallel partial decoder. Finally, precise edge detection and reverse attention were used to enhance and model the boundaries. The model produced decent segmentation scores, with an alignment index of 0.89 and a Dice of 0.74.

Using radiographs, the study [18] built a deep network called COVID-net in May 2020 to tell the difference between people with a SARS-CoV-2 infection and healthy people. Using the same image datasets, the proposed work was compared to ResNet-50 and VGG-19, the standard deep learning networks already trained. The suggested work does better than the VGG-19 and ResNet-50 models, with a positive predictive value (PPV) of 90% for healthy individuals, 91% for pneumonia, and 99% for COVID-19. A separate study [23] found that a deep COVID-XNet model could detect COVID-19 infection in 50 chest X-rays. The author utilised seven deep neural networks to extract image features successfully. They comprehensively compare the proposed model to other DL networks (DenseNet201 and VGG-19), and the outcome suggests that the approach has outstanding diagnostic performance with 90% accuracy.

The research in [24] developed a three-step DL model for classifying CT scan samples to find SARS-CoV-2 and identify it. This work uses different pre-trained models, such as SqueezeNet, ResNet101, ResNet50, and ResNet18, for transfer learning, abnormality localisation, and data augmentation. Experiments showed that the Resnet18 network model gave the best diagnostic performance, with accuracies of 99%, 97%, and 99% on the train, valid, and test sets, respectively. In another study [25], five network models (ResNet152, InceptionV3, ResNet101, Inception-ResNetV2, and ResNet50) were made to find COVID-19-infected patients using radiograph images. Cross-validation schemes were used to check this work by putting it into different classes for several binary classification tasks. The ResNet50 model was thought to be the best at finding coronavirus-2 because it had an overall accuracy of 98%. In [26], the authors suggested using artificial intelligence to come up with a fast way to find coronaviruses. The dataset had 1020 CT scan images from 194 patients, some of whom were healthy and some had COVID-19. The author used 10 CNNs to find COVID-19-positive people: VGG-19, VGG-16, ResNet-18, AlexNet, ResNet50, ResNet101, GoogleNet, Xception, SqueezeNet, and MobileNet. The authors reported that Xception and ResNet-101 attained excellent diagnostic performance among all networks. The ResNet101 model achieved an overall accuracy of 99%, while the Xception model achieved an overall accuracy of 100%. However, the work reported indications that the radiologists’ performances in detecting SARS-CoV-2 was modest. The author chose the ResNet101 network model as the best for screening and diagnosing coronaviruses. Therefore, it could be used in diagnostic applications. One recent work, in [27], developed five different deep learning models: ResNet50, ResNet101, DenseNet121, DenseNet169, and InceptionV3, for COVID-19 detection using chest X-ray images. The performance of these five models was evaluated using a large, publicly available library comprising CXR pictures associated with COVID-19 patients, as well as unknown data that had not before been seen by any model during the training or validation phase. The five studied DL algorithms produced good results, with the Resnet 101 model achieving the best accuracy of 96%, making it suitable for medical usage scenarios including the detection of COVID-19 cases using CXR Images.

Mahdy et al. [28] provided a complete scheme for recognising COVID-19 radiograph images. A support vector machine extracts in-depth features to categorise coronavirus-affected chest X-ray samples. This system is proposed as a beneficial Computer-Aided Diagnosis (CAD) tool to aid healthcare experts and medical doctors recognise COVID-19-positive cases early. This scheme reported excellent results in the categorisation of COVID-19. This work obtained an overall average accuracy of 97%. Samy et al. [29] recently presented a new CAD framework based on GLCM features with a latent-dynamics conditional random field (LDCRF) classifier model to classify COVID-19 positive and negative cases. This study extracted all ten features from GLCM for COVID-19 and non-COVID-19 chest X-ray images. The experimental result showed that the proposed method obtained an F1-score, recall, precision, and average accuracy of 95.5%, 94.6%, 96.1%, and 95.8%, respectively. However, this work is computationally complex because of the complex pre-processing and the extraction of several irrelevant features from CXR images. Recently, Oguz et al. [30] proposed a novel hybrid model for COVID-19 detection based on the combination of ResNet-50 and SVM. When tested on real data sets obtained from one hospital environment, the accuracy achieved by the proposed hybrid model (ResNet-50 + SVM) was 96.296%, F1 score of 95.868%, and an AUC value of 0.9821, which is greater than that of conventional ResNet-50 models alone.

This study in [31] presented a fuzzy logic-based deep learning (DL) strategy for distinguishing between Chest X-ray (CXR) images of patients with COVID-19 pneumonia and those with non-COVID-19 interstitial cases of pneumonia. The developed model, dubbed ‘CovNNet’, was used for extracting relevant features from CXRs combined with fuzzy images generated by a fuzzy edge detection algorithm. The experiments showed that using both the CXR and fuzzy feature inputs within a deep learning framework resulted in an accuracy rate of up to 81%, which was higher than benchmark DL approaches. Another work [14] proposed a Deep Learning Method (DLM) to detect COVID-19 using chest X-ray images. It was evaluated on 10,040 samples and had an accuracy rate of 96.43%, with a sensitivity of 93.68%, for correctly diagnosing COVID-19 instances compared to other expensive and time-consuming pathological tests, such as PCR or antigen testing kits, etc. Another recent work in [32] combined the deep learning model DenseNet 169 and Machine Learning model XgBoost to diagnose COVID-19. It achieved 98.23% accuracy in two-class problems and 89.70% accuracy in three-class problems, with 99.78%, 100% specificity and 92.08%, 95.20 sensitivity values, which are greater than other systems used before for detection purposes, such as the DarkCovidNet network. Table 1 demonstrates the related work performance along with its limitations.

## 3. Materials and Methods

### 3.1. Data Acquisition

We used a public dataset that Tawsifur Rehman [33] shared with us to test how well our proposed method works. This dataset, COVID-19, which a group of researchers from Qatar, Bangladesh, Malaysia, and Pakistan created in collaboration with medical professionals, was the winner of the COVID-19 database contest that the Kaggle community held. The database includes 3816 COVID-19 images, 345 pneumonia images, and 192 normal chest X-rays. In this study, we evaluated our proposed CAD system utilising a total of 1095 chest X-rays, including 375 normal images, 345 viral pneumonia images, and 375 COVID-19 images. Figure 2 displays image samples of normal and COVID-19. Whereas, in Table 2, the total images of each class are represented.

### 3.2. Proposed Methodology

This section covers the proposed strategy for automatic COVID-19 and pneumonia recognition. Below, Figure 3 shows the flowchart of our proposed framework. The overall structure of the proposed framework is as follows: First, each chest X-ray image is enhanced using a logarithm operator and Gaussian filter. Afterwards, hybrid features are extracted based on the Convolutional Vision Transform (CVT) and some statistical texture features by using the GLCM of chest X-ray images. These hybrid features are converted into a 1D vector representation and input into a random forest machine learning classifier to categorise images into three classes, namely, COVID-19, pneumonia, and normal. The proposed methodology’s workflow phases are explained in depth in the following subsections. Those steps are also described in Algorithm 1.

The approach used for implementing the proposed model is discussed below:
**Algorithm 1:** Classification of chest disease by using hybrid features and random forest classifier**Step 1:***Preprocess image, i.e., image = X and Preprocessing step are applied by using:**(a) Reshape image (X) to (500, 500)**(b) Remove noise using the Gaussian smoothing operator, and**(c) Enhance local contrast, logarithmic operator***Step 2:***Apply data augmentation technique on pre-processed images***Step 3:***Extract Hybrid Feature:**(a) Feature Extraction used: Next optimal texture feature of GLCM is used, which includes Contrast, Energy, Entropy, and Correlation, and we keep the distance 1 and 5, angle = 0 to reduce the computational complexity in feature contraction using GLCM.**(b) Afterward Vision transformer model is applied to extract more detailed spatial local features.***Step 4:***Hybrid-features = Both types of the above feature were combined and used as input to classifier***Step 5:***The classification used the output of previous feature extraction map steps, and a Random classifier is used to classify the images into three classes, namely, COVID-19, Pneumonia and Normal*

#### 3.2.1. Pre-Processing

The pre-processing step aims to detect and reduce the number of imperfections in the image. This phase is necessary for X-ray images since many radiological images contain noise and unwanted artefacts, such as patient clothing and wiring, that must be removed to identify COVID-19 accurately. In our proposed method, we apply simple pre-processing steps, which include image resizing, removing noise, and enhancing local contrast. We removed the noise from X-ray images using a Gaussian smoothing filter. Afterwards, a logarithm operator was employed to improve the local contrast of an image. This logarithm operator is also called the pixel logarithm operator because it enhances the low-intensity pixel value. Figure 4 represents the raw and enhanced image.

#### 3.2.2. Data Augmentation to Control Class Imbalance

Geometric augmentations are usually applied in combination to generate new augmented X-ray images. In this study, we evaluated our proposed CAD system utilised in a total of 1095 chest X-rays, including 375 normal, 345 viral pneumonia, and 375 COVID-19 images. First, we applied the percentage of data in the validation, testing, and training sets with the values of 10%, 20%, and 80%, respectively. Then, to avoid class imbalance, we applied a data augmentation technique. After applying the data augmentation technique, the 1095-image dataset is converted into 36,000 X-ray images, including 12,000 of normal, 12,000 of pneumonia, and 12,000 of COVID-19, as shown in Table 2. Figure 5 shows geometric augmentations that are applied individually to the X-ray image. This visualises the impact of each geometric augmentation method and gives the reader an idea about their relevance. Figure 5 demonstrates 12 different augmentation methods. From left–right and top–down, these are translation in the x-axis with +10 pixels, translation in the x-axis with −10 pixels, translation in the y-axis with +10 pixels, translation in the y-axis with −10 pixels, random shear in the x-axis within the range [−30, 30], random shear in y-axis within the range [−30, 30], random rotation within the range [−90, 90], random rotation within the range [−15, 15], horizontal reflection (or flipping), vertical reflection (or flipping), scaling in x-axis [0.85, 1.15], and scaling in y-axis [0.85, 1.15].

#### 3.2.3. Hybrid Features Extraction

After pre-processing the image, the next step is the extraction of hybrid features. The purpose of feature extraction is to extract discriminant features from imaging data. These features are then provided as input vectors to learning paradigms so that the machine can automatically generate a visual representation, analysis, or comprehension of image contents. This proposed work extracted the best discriminative statistical texture features of GLCM from chest X-ray images. These features included contrast, energy, correlation, and entropy. A second-order method was employed to obtain statistical texture attributes, which consider the connection of cluster pixels in the chest X-ray image I as input. The method has been employed in a variety of applications.

The GLCM P∈ℕN×N can be built as a frequency matrix by calculating the time every couple of quantised grey levels appear as neighbours in the quantified image QI. In more technical terms, each component of the GLCM P(i,j) can be easily figured as follows:(1)Pij∑k=1K∑t=1T{1, if QI(k,t)=i,QI(k+Δx,t+Δy)=j,0,  Otherwise,0
where δ=(Δx,Δy) denotes the displacement vector in pixel units along the x- and y-axes. It should be noted that a GLCM feature vector can be constructed using multiple displacement vectors. For example,
(2)$$\begin{array}{c} \Delta_{0^{{}^{\circ}}}=\left(1,0\right)\\ \Delta_{45^{{}^{\circ}}}=\left(1,1\right)\\ \Delta_{90^{{}^{\circ}}}=\left(0,0\right)\\ \Delta_{135^{{}^{\circ}}}=\left(-1,-1\right) \end{array}$$

If we take the transpose of the feature vector, the equation becomes
(3)$$\begin{array}{c} \Delta_{180^{{}^{\circ}}}=\left(-1,0\right)\\ \Delta_{225^{{}^{\circ}}}=\left(-1,-1\right)\\ \Delta_{270^{{}^{\circ}}}=\left(0,-1\right)\\ \Delta_{370^{{}^{\circ}}}=\left(1,-1\right) \end{array}$$

Discriminative information from input images is obtained in different directions and distances using GLCM to acquire the frequency of comparable patterns in multiple angles. Figure 6 [34] illustrates a GLCM computation. The first matrix is called the matric transformation matrix, whereas the second is the host image. Suppose there is a pair of (2, 2) pixels in the host image. If we consider distance one and angle zero, we find three positions in the matrix highlighted in red. We can write three for this pixel pair in the transformation matrix. This way, we can also generate other pixel pairs.

In this study, we used a distance of 1 and 5 and an angle of 0 to reduce computational complexity by applying an optimal collection of features. They are described below:

Contrast: This attribute determines the intensity values in an image at the local level. Contrast estimates the intensity contrast between neighbouring pixels over the entire image. Consequently, a low-contrast image has a smooth range of greys, but a high-contrast image has richer colours at both ends of the scale (white and black). It indicates that low spatial frequencies define a low-contrast image instead of low grey levels. Therefore, the GLCM contrast is closely correlated with spatial frequencies.

Energy: It denotes the homogeneity of grey distribution in an image. It can be defined as the quadratic sum of GLCM attributes.

Entropy: Another significant GLCM attribute to discriminate an image texture is entropy, a standard quantity of unpredictability typically regarded as a first-degree assessment of an image’s disorder level. The entropy obtained from the GLCM is inversely related to the energy feature of the GLCM.

Correlation: The GLCM correlation feature has comparable discriminative power to the contrast attribute. It delivers a numerical value that indicates how closely a pixel is connected or correlated with its neighbour throughout the image. It is defined as the linear grey-level dependence between pixels at specific locations concerning one another. Table 3 shows the numerical value of seven extracted features of some images.

The data was cleaned as necessary after being received from the Kaggle repository [33]. To execute a deep learning approach and obtain trustworthy results, a sizable amount of data is needed. However, every issue probably lacks sufficient evidence, particularly those that are medically connected. Medical data collection can be time- and money-consuming at times. Augmentation can be used to overcome these types of problems. A suggested model’s accuracy can be improved, and the overfitting issue can be solved by augmentation.

In addition, augmentation is used in this gathered dataset to avoid over-fitting. Rotation, zooming, and image sharing were among the augmentations. The data were then rearranged in order to make the model more inclusive and less overfitting. Following that, the suggested model was trained using the prepared dataset. To improve analysis, three distinct models were put into practice, and the accuracy of each was determined by comparing how well they performed. We used a novel approach, replacing the ReLU activation function in the supplied models with LeakyReLU activation. This procedure expedites training and prevents the issue of dead neurons (i.e., the ReLU neurons become inactive due to zero slope). The proposed model for chest X-ray image analysis is shown in Figure 1.

The Vision Transformer (ViT) [35] is likely the first completely transformer-based vision architecture, considering image patches as simple word sequences that are then encoded using a transformer. When pretrained on large datasets, the ViT can deliver outstanding image recognition results. However, without considerable pre-training, ViT performs badly in image identification. This is due to the transformer’s high model capability and lack of inductive bias, which leads to overfitting. Several subsequent studies have focused on sparse transformer models developed for visual tasks, such as local attention, to regularise the model’s capacity and improve its scalability successfully. The Swin transformer is an effective attempt to modify transformers by applying self-attention to shifting, non-overlapping windows. This methodology outperformed ConvNets on the ImageNet test for the first time using a pure vision transformer. Window-based attention was discovered to have limited model capacity due to the loss of non-locality, and, hence, scales badly on larger data sets, such as ImageNet-21K, despite being more adaptable and generalisable than the complete attention used in ViT. However, because the attention operator has quadratic complexity, full-attention acquisition of global interactions in a hierarchical network at early or high-resolution stages requires computationally significant effort. It is still difficult to include global and local interactions while maintaining model capacity and generalizability within a computer cost.

Shift, scale, and distortion invariance are Convolutional Neural Networks (CNNs) aspects. These aspects are translated to the ViT architecture [36] while the benefits of transformers have been retained. (i.e., dynamic attention, global context, and better generalisation). Although vision transformers are effective on a broad scale, they perform worse when trained on less data than smaller CNN rivals (such as ResNet). One argument might be that because CNNs naturally exhibit some desired features ViT lacks, they are better suited to addressing vision-related concerns. A texture forces the capture of this local structure by using local receptive fields, shared weights, and spatial subsampling. As a result, it achieves some shift, scale, and distortion invariance. For instance, images frequently have a strong 2D local structure with closely spaced-apart pixels intimately related. Additionally, learning various complex visual patterns, from low-level edges and textures to higher-order semantic patterns that account for local spatial context, is made possible by the hierarchical structure of convolutional kernels.

The convolutional projection is the first layer of the convolutional transformer block. In this study, we proposed that, while maintaining a high level of computational and memory efficiency, convolutions might be selectively introduced to the ViT structure to enhance performance and robustness. In our work, we only incorporated convolutions blocks from transformer, and was innately efficient in terms of parameters and floating-point operations, which was given as evidence supporting our idea. We added an Attentional Selective Fusion (ATTSF) layer to CvT to emphasise the local and global interactions of pixels. Token embedding and projection for attention hierarchical transformers were integrated into the original CvT model by the authors as part of a sophisticated strategy. However, we adopted a computationally effective strategy via an ATTSF mechanism.

In our work, the proposed convolution block to act as Feature Learning (FL) to extract robust feature from CXR images. The FL extracts learnable features from CXR images. A group of convolutional procedures makes up the feature learning (FL) component. The FL component follows the hierarchy of the Inception v3 architecture. The FL component differs from the Inceptionv3 model in that it does not contain the Fully-Connected Layer (FCL) present in the Inceptionv3 design and instead extracts features for our classification component. A CNN without the FCL layer is the outcome.

Additionally, we included an attentional selective fusion (ATTSF) that combines global and local attention to add flexibility when combining different types of information. To obtain more local and global interactions, we initially fused the best GLCM texture features with the CNN features (CNN) using only the feature learning component of CVT in this work. Afterwards, fused features were supplied into the random forest, which converts them into COVID− and COVID+ for the final detection stage.

As shown in Figure 1, our attentional selective fusion (ATTSF) included both local and global attention, which could increase flexibility when combining different types of information. As previously said, to capture the following information exchange, we first fused the best GLCM features (contrast, energy, entropy, correlation) and the CNN features, yielding two feature maps ($GLC_{cont}$, $GLC_{eng}$*,*
$GLC_{ent}$*,*
$GLC_{corr}$*, CNN*) acquired from the backbones:(4)$$Features{\left(i,j\right)}_{map}=GLC_{cont}+GLC_{eng}+GLC_{ent}+GLC_{corr}+W\times CNN\;$$

In the above equation, the extracted feature map uses GLCM features and CNN is the integrated feature map. It was simple to generate the W weights for the initial integration using two 1 × 1 convolutions. Next, we used these fused feature maps for classification tasks using a random forest classifier.

#### 3.2.4. Formulation of Classification Model

The feature classification part of our proposed system distinguishes SARS-CoV-2 and non-SARS-CoV-2 instances, as described in this section. This step in our work primarily categorises every chest X-ray image into two diagnostic classes (COVID-19 and pneumonia normal) based on texture information obtained. The classification module relies heavily on the availability of clinical diagnostic cases. This collection of instances with an early diagnosis is referred to as the “training set”. The employed learning strategy is known as “supervised learning”. Various classification strategies are available in the literature for the recent job of COVID-19 classification, including Bayesian networks, SVM, conditional random fields, latent-dynamic CRF, k-nearest neighbour, artificial neural networks, etc. We used a random forest machine learning classifier for the classification task in this work. Random Forest (RF) [37] is an ensemble technique proposed by Breiman for classification problems. This machine-learning approach boosts the system’s accuracy by combining several models to solve the said problem. The participation of several decision models usually results in inaccurate predictions compared to the prediction obtained using a single model. This model is the best machine-learning algorithm for classification problems in different research domains [38,39,40] because it can draw its training data from randomly selected subsets and construct trees in a similarly haphazard fashion [41]. Recent studies show that random forest classifiers present encouraging results in several healthcare systems [39,42]. Figure 7 represents the operation flow of the random forest classifier. The parameters used in the random forest algorithm are described in Table 4.

The random forest classifier linked with features and training samples $S_{n}$ could be depicted as:

Beginning: $S_{1},S_{2},S_{3},\ldots ,S_{n}$ were sampled with replacement using a preset probability.For each sample, $S_{n}$, build a decision tree. The training samples are chosen randomly from the available features using the subspace of the m-try dimension. Compute all the possibilities regarding all features. The leaf node yields the best data split. The operation will be repeated until the saturation threshold is met. We investigated hyper-parameters and selected the optimal parameter for our proposed CAD system to improve the algorithm’s performance. Table 4, below, shows the hyper-parameter for the RF classifier.A combination of unpruned trees $h_{1}(X_{1}),h_{2}(X_{2}),\ldots$ of N number in a random forest ensemble uses the highest possible value for classification decisions.

## 4. Results

The proposed automated detection method was developed and applied in Python using the Anaconda framework and OpenCV Library. This vision library is used for real-time object detection and digital image processing. All experiments, analyses, and evaluations were performed on a Dell laptop equipped with an Intel(R) Core (TM) i7 CPU, 16 GB of RAM, a 1.50 GHz processor, and a Microsoft Windows 10 × 64 version.

### 4.1. Performance Evaluation Metrics

Several assessment measures, such as sensitivity, specificity, F1-score, accuracy, recall, and precision, are used to quantitatively evaluate the performance of the proposed technique. Four key metrics are utilised to calculate these measures: (a) correctly identified unhealthy cases (*TP*), (b) incorrectly diagnosed diseased cases (*FN*), (c) correctly identified healthy cases (*TN*), and (d) incorrectly classified healthy cases (False Positives, *FP*).

*Accuracy* (ACC): The most often used, and one of the basic, performance measures is accuracy, which is the probability that a randomly selected example (negative or positive) would be accurate. The diagnostic test in this metric shows the probability of a correct result, i.e., the probability that the diagnosis will be correct.
(5)$$Accuracy=\frac{TP+TN}{TP+TN+FP+FN}$$

Precision (*PR*): It relates to the ability to detect positive classes among all accurately predicted positive classes, represented as a proportion of all correctly predicted positive classes to all correctly predicted positive classes.
(6)$$PR=\frac{TP}{TP+FP}$$

Sensitivity (*SEN*): Also known as recall, true positive rate, or hit rate, is a measure of a model’s capability to detect all positive cases. It is worth noting that the above equation implies that a low false-negative rate almost always accompanies a high recall.
(7)$$Recall\;or\;SEN=\frac{TP}{TP+FN}$$

Specificity (*SPE*): The data’s ratio of true negatives to total negatives.
(8)$$SPE=\frac{TN}{TN+FP}$$

F1-score: It is not as straightforward as accuracy, but this metric is useful in determining the classifier’s exactness and robustness. The F1 score, a fundamental test performance statistic that considers both memory and precision, is typically computed as a weighted average of recall and precision.

The *TN* (true negative) and *TP* (true positive) are accurately predicted negative and positive outcomes, respectively. *FN* (False Negative) and *FP* (false positive) are not correctly predicting negative and positive COVID-19 cases.

Area Under the Curve (AUC): It stands for “Area Under the Curve.” It gives the performance measure on all thresholds. The AUC scale runs from 0 to 1, with 0 being the lowest and 1 being the highest. The AUC of a model with 100% inaccurate predictions is 0.0, whereas the AUC of a model with 100% accurate predictions is 1.0. Figure 8 depicts the AUC representation.

### 4.2. Results Analysis

Three experiments were performed in this work. The first experiment was about selecting the best possible combination of texture features for our computer-aided diagnostic (CAD) system. The second experiment evaluated our proposed CAD using different train–test splits with different performance metrics. The third experiment was about the performance analysis of the proposed method compared to other state-of-the-art studies found in the literature.

In experiment 1, we first evaluate the proposed classifier’s accuracy with each texture attribute, and these single features offer different accuracy results. Here, for this experiment, we have evaluated the classification accuracy on three different train–test splits, i.e., 90–10%, 80–20%, and 70–30%, to select the best combination of features. A total of six texture features, such as energy, correlation, dissimilarity, homogeneity, contrast, and entropy, were evaluated. The accuracy of a random forest classifier using various texture features was investigated in Table 5. We have noticed that the classification accuracy of using a single feature, namely energy, correlation, contrast, and entropy, reported more than or equal to 85% on three train–test splits. It shows the potential of these features to detect normal, pneumonia, and COVID-19 chest X-ray images. In the next experiment, we have combined those best four features to investigate further the potency of the proposed work using different train–test splits.

In this experiment, we select the four best features from experiment 1. We combined them and extracted the best four features (energy, correlation, contrast, and entropy). We used them with a random forest classifier to classify chest X-ray images into COVID-19 and normal images. Three train–test splits (90–10%, 80–20%, and 70–30%) were used to better evaluate the proposed method’s performance. We have noticed from Table 5 that we obtained a good detection accuracy on each train–test split and an average accuracy of 97%. We can also observe from the table below that our proposed method performs well and obtains an F1-score value of greater than 93% on all train–test splits. It shows our proposed scheme’s significance and effectiveness in detecting COVID-19 and normal cases. In addition, our model obtained good precision, also known as positive predictive value (100%, 99%, and 95%, respectively, for COVID-19 patients and 100%, 91%, and 91% for normal patients), and recall (100%, 91%, and 90% for COVID-19 cases, and 100%, 99%, and 96% for normal cases), which shows the supremacy of the proposed work.

Figure 9, below, represents our proposed work’s confusion matrix using different train–test splits. There are a lot of interesting observations that can be made regarding how our proposed method operates in various settings. First, our proposed state-of-the-art algorithm has a high sensitivity for COVID-19 cases (100%, 91%, and 91%, respectively, under different train–test splits), as shown in Table 5 and Figure 9, below, which is significant since we aim to minimise the number of missing COVID-19 instances. Second, the detection of negative cases is also high; we achieved a specificity of 100%, 99%, and 94% in different split settings. As a result of these findings, it is clear that the proposed feature and the machine learning framework-based CAD system as a whole do a great job of recognising COVID-19 instances from CXR images. In addition, Figure 10 demonstrates the comparison graphs for model accuracy with loss related to both data augmentations and no augmentation techniques on training and testing datasets. These graphs show that the data augmentation technique is very important in solving the issue of data imbalance.

Figure 11 shows the AUC representation of our proposed classifier using three different training–test splits. Figure 11 shows that our proposed model predicts nearly 100% of the time on three different training–test splits. The AUC evaluates the proposed classifier’s outstanding classification performance by comparing the true and false positive rates when identifying COVID-19 instances from test images (a chest X-ray). Hence, this satisfies the fact that the proposed work is recommended and would be an efficient choice for detecting COVID-19 cases.

As was previously said, it is crucial to determine how the network picks up information from the CXR pictures. Score-CAM-based heat maps made from original (not segmented) and segmented CXR images can be used to make sure that the classification algorithm can learn what parts of the CXR images are important and what parts are not.

The heat map of the segmented lungs and the original lungs shows that the CNN models’ decision-making in the original CXR does not always originate from the lung sections (Figure 12). When CNN uses plain X-ray pictures to put things into groups, itis not always the lungs that are the most important parts of the CXR images. However, it is clear from Figure 12 that the proposed CAD system can work well with X-ray images, which is good for a biomedical application as important as this one. On the other hand, segmented CXR images are better for computer-aided diagnosis because they make it easier to put diseases into the right categories using chest X-ray images.

The proposed diagnostic system was compared to similar research [25,34,35,36,37,38] conducted in the past. This was performed to show that the proposed diagnostic system is better than the current detection schemes regarding several estimation metrics. Table 6 summarises the details of the quantitative comparison. Our proposed CAD model, based on three unique train–test splits, produces a good accomplishment regarding the F1-score, precision, recall, and average accuracy with 96%, 96%, 96%, and 97%, respectively. It can be noted from Table 6 that our proposed approach outperforms the other similar studies in terms of recall, precision, F1-score, and average accuracy. Therefore, our proposed CAD scheme could be a significant addition to healthcare centres to help diagnose COVID-19 cases.

From pre-processing the image to the last step of the proposed method, the presented CAD solution for COVID-19 takes about 2 s to figure out, on average. This means that it can be used for real-time processing. Moreover, the proposed work has low computational costs for pre-processing steps, real-time texture feature extraction, and classification.

## 5. Discussion

The first case of the fatal infectious illness coronavirus disease 2019 (COVID-19) was discovered in Wuhan, China, in December 2019 [28,38,48]. A huge epidemic has been caused by the COVID-19 virus because there has yet to be a cure found for it. Because the disease changes over time and has a structure with only one strand of RNA, it is hard to treat. Thousands of people have died because of COVID-19, which has notably affected nations such as the United States, Spain, India, Italy, China, the United Kingdom, Iran, etc. Humans, cats, dogs, pigs, chickens, rodents, and other animals all carry different strains of COVID-19. Symptoms of COVID-19 include sore throat, fever, headache, nose flushing, and cough. People with compromised immune systems are most vulnerable to the virus, which can be fatal. This contagious COVID-19 illness spreads quickly from person to person around the world. The major ways that this may transfer from one person to another include physical contact, breath contact, hand contact, or mucous contact. A virus family that includes this one is known to cause severe respiratory problems. There are spikes on the crown and outer surface of the virus’s structure. Middle East respiratory syndrome (MERS) and severe acute respiratory syndrome (SARS) are also included in this group [45]. Severe lung damage and acute respiratory discomfort are additional effects of such circumstances [10]. As of 15 July 2020, there were 12,964,899 afflicted individuals around the planet, resulting in 570,288 fatalities. The current situation indicates that people with chronic health conditions and the elderly appear to be more vulnerable to the COVID-19 death rate. The virus is passed from person to person by coughing, sneezing, and respiratory droplets [44]. Fever, inflammation, abnormalities in the respiratory system, and illnesses including pneumonia, multiple organ failure, and death are common symptoms of this virus [36,41]. Laboratory tests are expensive, time-consuming testing methods that call for a well-equipped research lab.

On CT scans, a deep learning-based algorithm is used to look for COVID-19. Several researchers have also produced and made chest X-ray images of COVID-19 patients accessible in public databases [15,16]. These open datasets are used to diagnose COVID-19 using a technique called COVID-Net [21]. Deep learning was used to diagnose from chest X-ray images, and the results were promising. For processing medical imaging data, deep learning models are frequently utilised. In Ref. [16], convolutional neural networks are used to identify pneumonia. This research proposes an automated technique for the deep network-based diagnosis of COVID-19. The suggested network makes use of the multiresolution analysis feature. There are several benefits to using deep networks and wavelet transformations together. The network receives the wavelet decomposition as input. A conventional convolutional neural network (CNN) is not being used. In this study, a depth-wise separable network is used.

An automated approach for detecting COVID-19 from chest X-ray pictures was given in the paper through the creation of an enhanced depth-wise convolution network with spectrum analysis. The convolution and pooling layers have been rewritten to be more general cases of filtering and downsampling. This reformulation incorporates depth-wise networks and multiresolution analysis. For multiresolution analysis, the input pictures are deconstructed using the Haar wavelet. Fix-weight filters are used to apply the wavelet. The developed model is used to detect COVID-19 illness on chest X-ray images. The photos are divided into three categories by the model: normal, viral pneumonia, and COVID-19. A comparison analysis is also performed to assess how well the suggested approach works. From chest X-ray pictures, the suggested approach may be utilised to diagnose COVID-19. The utilisation of X-ray pictures will aid in illness management.

### 5.1. Advantages of the Current Study

Our extensive research project is focused on utilising CXR images to diagnose COVID-19. For identifying normal X-rays and COVID-19 images, our effort involves pre-processing, feature extraction, and classification stages. The contributions of our suggested work are listed below.

(1)We have created a novel feature framework based on a convolution vision transformer and an optical set of GLCM characteristics, such as contrast, energy, entropy, and correlation, to extract attractive features from improved chest X-ray pictures.(2)The Gaussian filter and the logarithmic operator were used to construct a low-cost and straightforward pre-processing approach.(3)The random forest classifier was used to classify traditional X-ray and COVID-19 pictures using three different train–test split strategies.(4)Accuracy, precision, recall, and F1 score were used as performance indicators to analyse the results. A comparison of the planned study with earlier related work for the detection of COVID-19 was also provided.

### 5.2. Limitations of the Current Study

Deep learning (DL) models, according to recent research, can learn from irrelevant data and make decisions based on that data [43,44,45,46], even though high-performing networks’ performances cannot be extrapolated to real-world applications. In contrast to ordinary X-rays, the segmented lungs helped the CNN model to identify the primary Region of Interest (ROI). In other words, the end-user’s trust in the performance of Artificial Intelligence (AI) must be increased by the consistency of the categorisation judgments made by the network. The lung area in the CXR images should be used to make conclusions about lung diseases instead of using whole images. The results are presented using ordinary X-rays, which can partially or entirely fail in a real-world application because there are not many ground-truth of using masks available these days. Several authors created a benchmark of lung masks with the aid of a group of radiologists. However, they are not sufficient to train the model.

In a sample case, almost all image enhancement techniques misclassified COVID-19 X-rays as normal or non-COVID lung opacity, but the gamma-correction-based image enhancement technique classified it correctly. It is interesting to see how the gamma enhancement technique outperforms other enhancement techniques. To appropriately identify the lung image, the gamma correction approach on the segmented lungs uses judgments made by the region of interest, or lungs, as shown in Figure 12. To summarise, COVID-19 and other lung infection detection performance reported in recent literature is comparable to the performance described in this study (Table 6). However, this study reports the crucial elements that are absent from other recent publications. In addition, no article has reported results using such large CXR images and corresponding ground truth lung masks. As a result, given that this study trained and validated its models on a sizable dataset, its results are comparable to those of the state-of-the-art, trustworthy, and generalisable.

## 6. Conclusions

This work proposes a cost-effective diagnostic scheme to identify COVID-19 cases using CXR images. The proposed method comprises three steps: image pre-processing, feature extraction, and classification. This work adopted simple pre-processing techniques based on the Gaussian filter and logarithmic operator to enhance image quality. Then a fused feature extraction scheme, based on a convolutional vision transformer and an optimal set of GLCM features, such as contrast, correlation, entropy, and energy, was applied, and the features were extracted from each enhanced chest X-ray image. Finally, the extracted features from images were input to a random forest classifier. The proposed method obtained excellent average accuracy, precision, recall, and precision of 97%, 96%, 96%, and 96%, respectively. The proposed scheme performs better than similar work in the literature using chest X-ray images. This study investigates how various image enhancement methods affect deep convolutional neural networks’ ability to automatically recognise COVID-19 from CXR pictures. A shallow CNN model was built from scratch in this paper, and six different deep learning pre-trained CNN models were trained with ImageNet weights. The performance of seven CNN models for five distinct image enhancement approaches was used to investigate the categorisation of COVID-19, non-COVID lung infection, and normal CXR images. Our thorough research on image enhancement methods demonstrates that an accurate COVID-19 diagnosis may be made with 96.29% accuracy, 96.28% precision, and 96.28% recall.

Also, because it uses a traditional feature extraction method and a traditional machine learning classifier, our proposed CAD system takes less time and money to compute. Therefore, this method could be a suitable choice for the real-time diagnosis of COVID-19. The proposed method can be trained and evaluated with a larger dataset in the future, and numerous other machine-learning methods can be included for performance evaluation.

## Figures and Tables

**Figure 1 healthcare-11-00837-f001:**
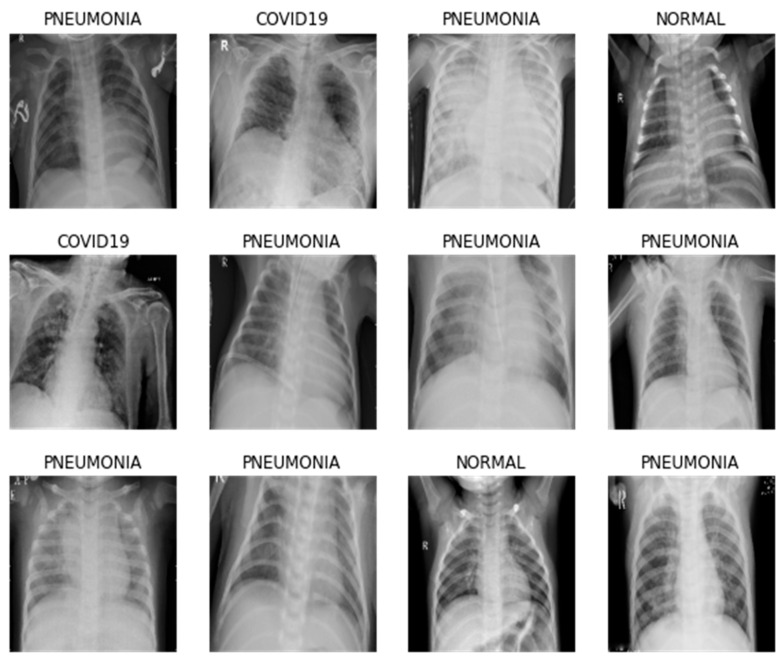
Examples of healthy, pneumonia, and COVID-19 chest X-ray images.

**Figure 2 healthcare-11-00837-f002:**
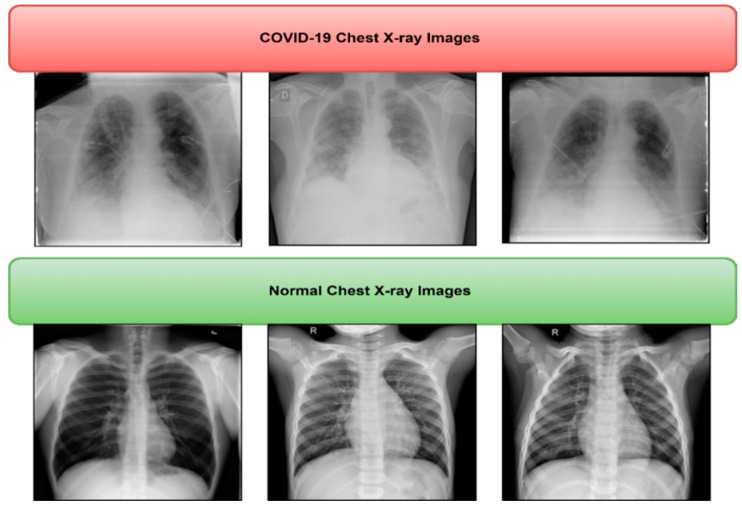
Sample of normal and COVID-19 radiograph images from Kaggle dataset.

**Figure 3 healthcare-11-00837-f003:**
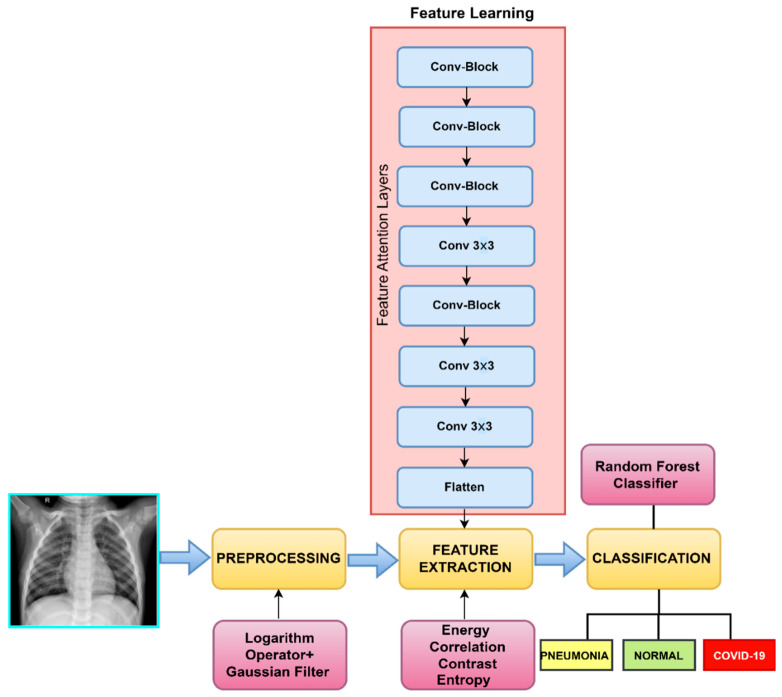
Representation of the steps involved in the proposed diagnostic method.

**Figure 4 healthcare-11-00837-f004:**
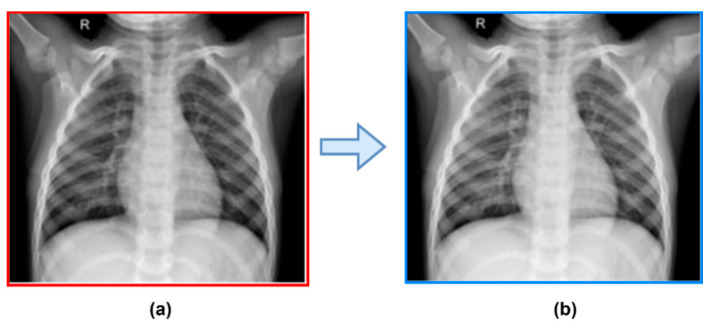
Contrast enhancement and smoothing image: (**a**) Raw image (**b**) Enhanced image.

**Figure 5 healthcare-11-00837-f005:**
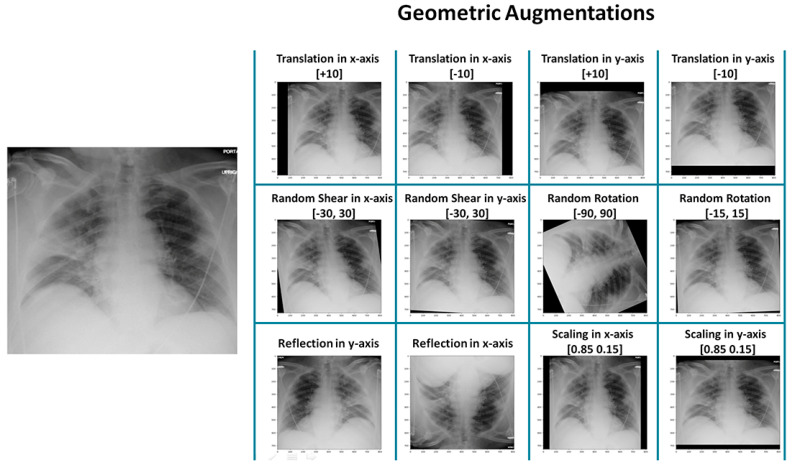
Some examples of data augmentation techniques to balance the classes.

**Figure 6 healthcare-11-00837-f006:**
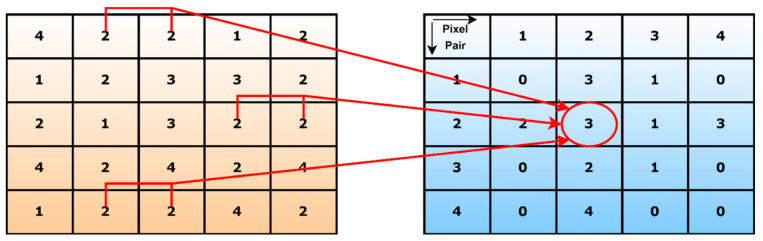
An example of GLCM computation.

**Figure 7 healthcare-11-00837-f007:**
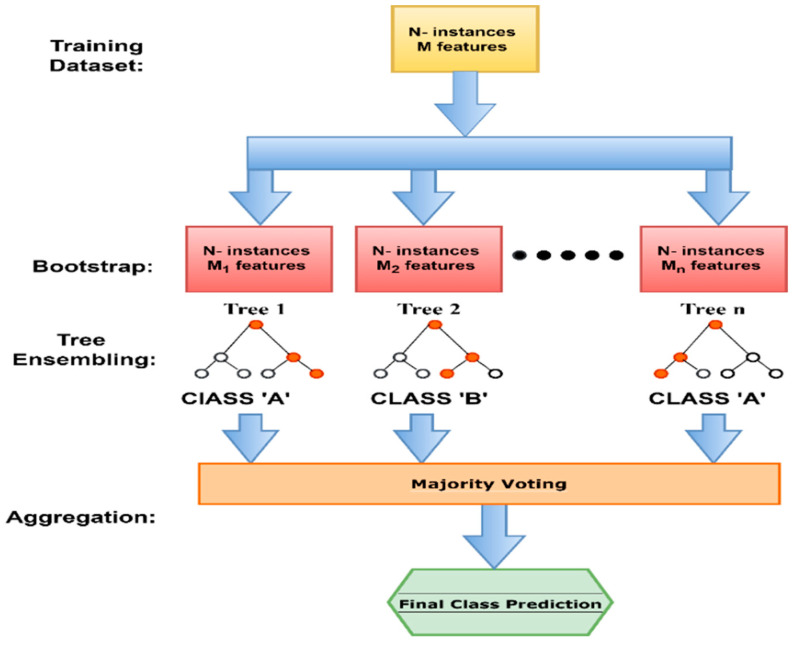
An example of the operation flow diagram of a random forest classifier.

**Figure 8 healthcare-11-00837-f008:**
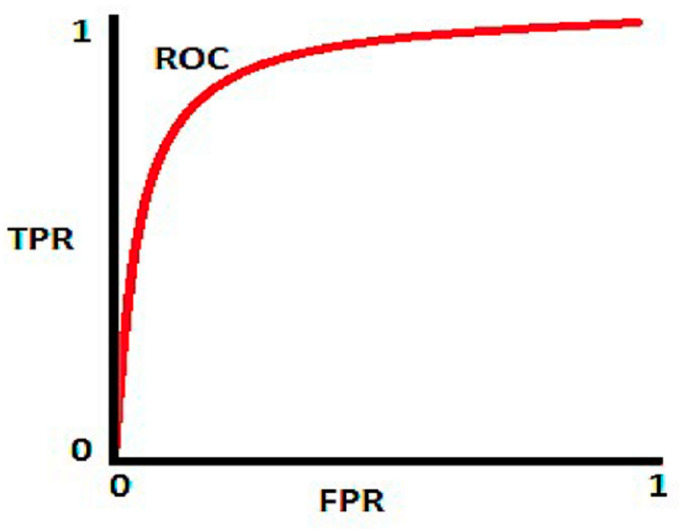
Illustration of area under the ROC curve.

**Figure 9 healthcare-11-00837-f009:**
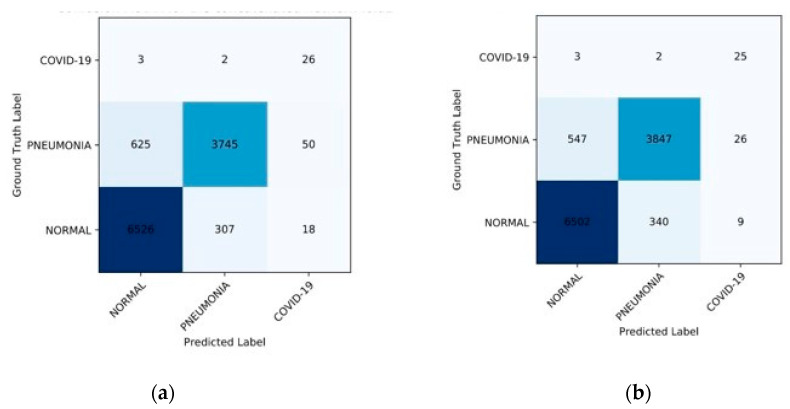
Confusion Matrix with different train–test splits: (**a**) Train–test split 80–20% and (**b**) Train–test split 70–30%.

**Figure 10 healthcare-11-00837-f010:**
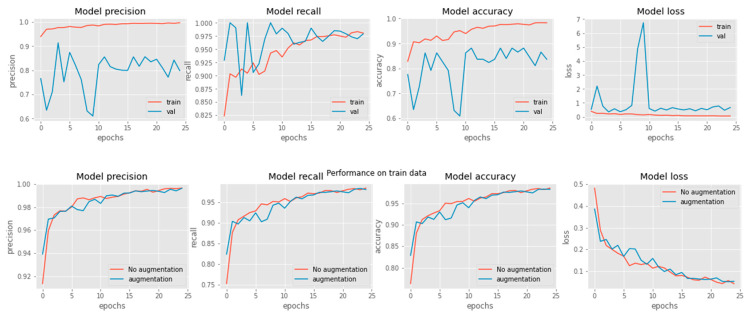
Comparisons graph for proposed model accuracy compared with loss related to data augmentations and no augmentation.

**Figure 11 healthcare-11-00837-f011:**
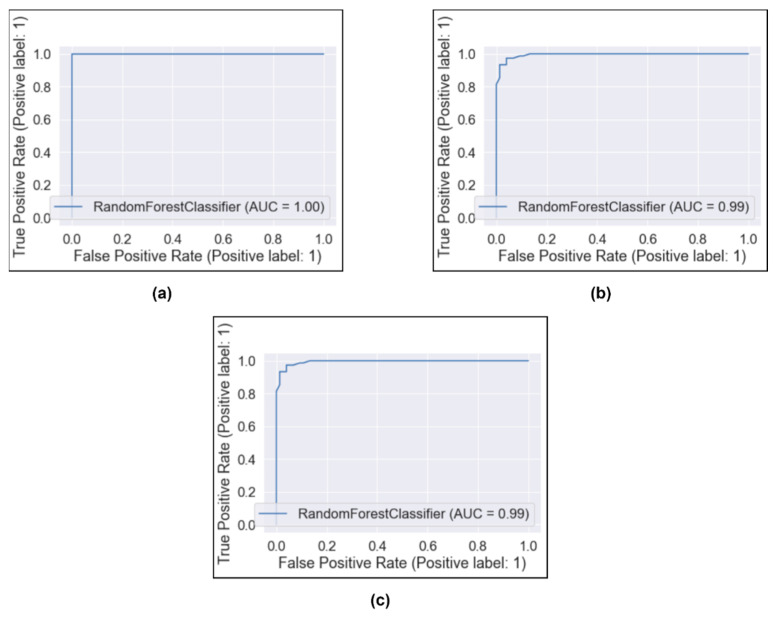
AUC with different splits: (**a**) Train–test split 90–10%, (**b**) Train–test 80–20%, (**c**) Train–test 70–30%.

**Figure 12 healthcare-11-00837-f012:**
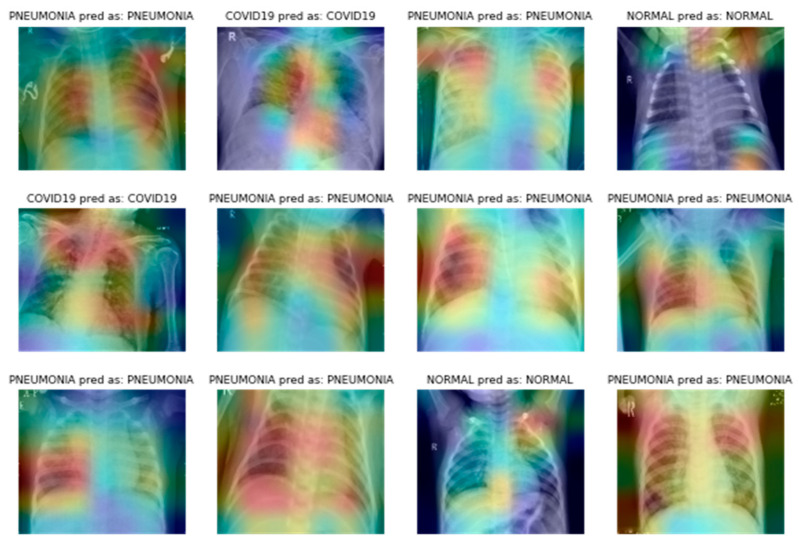
The segmented CXR image heat maps show the proposed models’ decision-making capabilities for different chest diseases.

**Table 1 healthcare-11-00837-t001:** Summary of related work for COVID-19 detection.

Cited	Methodology	Dataset	Results	Parameters	Limitations
Yujin et al. [12]	A Patch-based CNN method was developed based on ResNet-18 architecture.	JSRT + NLM	ACC = 88%, F1-score = 84%, SEN = 96%, SPE = 81%	NIL	The method is considered computationally complex because of complex image pre-processing and classification steps.
Tulin et al. [21]	The DCN model was proposed.	OWN Datasets	ACC = 98%, F1-score = 96%, SEN = 95%, SPE = 95%	678 Trainable	The limited number of COVID-19 samples used.
Fan et al. [22]	The inf-Net deep learning model was developed.	COVID-19 CT Collection	SEN = 87%, SPE = 97%.	NIL	Only focused on lung infection segmentation rather than classifying COVID-19 patients. Complex approach to obtain multi-class label infection label.
Linda et al. [18]	The covid-Net deep learning model for detection.	COVIDx	ACC = 93%	11.78 million	The method has degraded performance.
Ezz et al. [23]	The covidXNet model was proposed	OWN Datasets	ACC = 90%	NIL	Training complexity; several deep learning models were used to train.
Sakshi et al. [21,24]	Several pre-trained models were used in the study.	CT scan Dataset	ACC = 99%. SEN = 100%, SPE = 98%	NIL	No effective evaluation of the deep learning model because only one train–test split strategy is used.
Ali et al. [25]	Several deep learning models are use in this work.	Dr Joseph Dataset, ChestX-ray8 Dataset, Kagge ChestXray Datasets	ACC = 96%, ACC = 99%, ACC = 99%	210.4 Millions	Several deep learning models used in the proposed work, making them complex.
Abbasian et al. [26]	A comparative study of 10 deep learning models was presented.	Own Datasets	ACC = 100%	NIL	Limited in terms of model complexity.
Lamia et al. [28]	A multi-level threshold and SVM method were proposed.	Montgomery County X-ray Set	ACC-97%, SPE = 99%, SEN = 95%	NIL	Low generalizability problem.
Constantinou et al. [27]	Several deep learning models are used in this work.	Chest X-ray Image	PRE = 96%. RECALL = 96% ACC = 96%	NIL	Low generalization problem.
Oguz et al. [30]	ResNet 50 and SVM Model.	CT Images	ACC = 96%, F1_score = 95%	23 Million	Limited in terms of data used in work
Ieracitano et al. [31]	A fuzzy logic-based deep learning (DL) approach called CovNNet	CXR Images	ACC = 81%	NIL	Degraded performance
Chakraborty et al. [14]	DLM method is used to detect COVID.	CXR Images	ACC = 96%. SEN = 93%	11 Million	Lack of model validation because of limited data.
Nasiri et al. [32]	DenseNet 169 and extreme gradient boosting	CXR Images	ACC = 98%.	NIL	Low sensitivity achieved in two-class problem

**Table 2 healthcare-11-00837-t002:** Number of images in each class of COVID-19 database, with data augmentation and without data augmentation.

Classes	No. of Images	Data Augmentation
normal	375	12,000
pneumonia	345	12,000
COVID-19	375	12,000
Total	1095	36,000

**Table 3 healthcare-11-00837-t003:** Extracted features of some images.

	Features						
Images	Energy_1	Correlation_1	Contrast_1	Energy_2	Correlation_2	Contrast_2	Entropy
Image_1	0.052535	0.994353	30.0105	0.0348948	0.936108	327.297	7.18441
Image_2	0.0267825	0.993088	22.6968	0.0158728	0.939816	194.681	7.24483
Image_3	0.0597171	0.992787	78.1387	0.0436013	0.923739	815.503	7.60572
Image_4	0.0270262	0.991886	37.3811	0.0158361	0.902464	440.749	7.32227
Image_5	0.0380929	0.995518	15.8684	0.0226321	0.965529	122.115	7.15369

**Table 4 healthcare-11-00837-t004:** Parameter used in random forest algorithm.

Parameters	Range
n_estimator	60
min_samples_split	2
max_depth	None
min_samples_leaf	1
Random_state	42

**Table 5 healthcare-11-00837-t005:** Classification accuracy with different train–test splits.

Features	Classifier	Train–Test Splits	Class	Precision	Recall	F1-Score	SEN	SPE	ACC	Avg. ACC
Energy + Correlation + Contrast + Entropy + Convolution Block	Random Forest	90–10%	COVID-19 Normal	100% 100%	100% 100%	100% 100%	100%	100%	100%	97%
		80–20%	COVID-19 Normal	99% 91%	91% 99%	95% 96%	91%	99%	96%	
		70–30%	COVID-19 Normal	95% 91%	90% 96%	93% 93%	91%	94%	93%	

**Table 6 healthcare-11-00837-t006:** Performance comparison of proposed work with earlier works.

Method	Recall	Precision	F1-Score	Accuracy
Podder et al. [43]	94%	94%	94%	94%
Panwar et al. [44]	82%	97%	89%	88%
Echtioui et al. [45]	86%	96%	91%	94%
Shukla et al. [46]	90%	92%	91%	87%
Kunar et al. [47]	83%	89%	82%	93%
Samy et al. [29]	94%	96%	95%	95%
Our Proposed CAD system	96%	96%	96%	97%

## Data Availability

We have used a publicly available dataset shared by Tawsifur Rehman [33]. This dataset can be downloaded at https://www.kaggle.com/tawsifurrahman/covid19-radiography-database (accessed on 22 December 2021).

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
