# Peer review of "Computer-Aided Diagnosis of COVID-19 from Chest X-ray Images Using Hybrid-Features and Random Forest Classifier"

_healthcare, 2023, doi:10.3390/healthcare11060837_

Round 1
Reviewer 1 Report
The paper addresses a widely researched area recently.
Abstract: The authors claim that “ A comparative study shows that our proposed method outperforms existing and similar work.”. However, many existing studies outperform these performance” metrics.
Related studies: It would be better to consider the latest related studies for chest x-ray classification. For example https://doi.org/10.1016/j.asoc.2022.109319
Methodology: considering the performance of the existing studies, I believe that the authors should refine their methodology to obtain higher results.
It would be better to include a table, that shows the number of images in each class, for each of the training: testing: validation sets; together with the original numbers and the number of images after augmentation (actual images fed to the model). Clearly indicate whether you have
- Perform augmentation and separate the dataset into train:test: validation (if you have used this, then the method is not correct, as the variations of the same image can be in all sets)
OR
- First split into train:test: validation and then perform augmentation
Results: consider the learning curves (graphs): eg. Figure 9. The 2nd row is for the training dataset. Please include the validation dataset-related results.
Mainly, you can include the training and validation-related curves (accuracy, loss) in the same chart, so we can observe whether the model is overfitting.
It if overfits, you need to train the model again, addressing those issues as well, to obtain better performance.
You have addressed explainability/ interpretability as well using heatmaps. How do you evaluate/ validate those results? It would be better to do a user study with a set of radiologists, for the correctness of the XAI results.
Discussion: Please compare the features and results of the proposed study with the latest existing studies. Highlight the novel contribution, compared to the related studies. Some of the latest studies can be listed as follows:
https://doi.org/10.3991/ijoe.v18i07.30807
Author Response
Response to Reviewer 1 Comments
The paper addresses a widely researched area recently.
- Abstract: The authors claim that “A comparative study shows that our proposed method outperforms existing and similar work.”. However, many existing studies outperform these performance” metrics.
Response: Thanks, dear reviewer, as per your suggestion, we have updated our abstract those sentences as:
“A comparative study with the existing methods is also performed. For performance evaluation, metrics such as accuracy, sensitivity, and F1-measure are calculated. The performance of the proposed method is better than that of the existing methodologies, and it can thus be used for the effective diagnosis of the disease.
”
- Related studies: It would be better to consider the latest related studies for chest x-ray classification. For example https://doi.org/10.1016/j.asoc.2022.109319
Response: Thanks dear reviewer for suggesting important study. Yes, we have revised and include the latest studies in the related section of our paper. The changes can be seen on page 4 in related section of the paper.
- Methodology: considering the performance of the existing studies, I believe that the authors should refine their methodology to obtain higher results.
It would be better to include a table, that shows the number of images in each class, for each of the training: testing: validation sets; together with the original numbers and the number of images after augmentation (actual images fed to the model). Clearly indicate whether you have
- Perform augmentation and separate the dataset into train:test: validation (if you have used this, then the method is not correct, as the variations of the same image can be in all sets)
OR
- First split into train: test: validation and then perform augmentation
Response: Thanks, dear reviewer for taking my attention to this mistake. Yes, you are right. We have first split and then add data augmentation technique. We have updated our paper according to it. On page#9, we have added the following updates
“
First, we applied the percentage of data in the validation, testing and training sets with the values of 10%, 20% and 80%, respectively. Then to avoid class imbalance, we have applied data augmentation technique. After applying data augmentation technique, the 1095 dataset is converted into 36,000 x-ray images including 12,000 of normal, 12000 of pneumonia and 12000 of covid-19 as shown in Table 2.
”
A new table 2 is also added on page# 10 to describe the images with and without data augmentation technique.
Table 2. Number of Images in each class of COVID-19 database with data augmentation and without data augmentation.
|
Classes* |
No. of Images |
Data augmentation |
|
normal |
375 |
12,000 |
|
pneumonia |
345 |
12,000 |
|
COVID-19 |
375 |
12,000 |
|
Total |
109,5 |
36,000 |
- Results: consider the learning curves (graphs): eg. Figure 9. The 2ndrow is for the training dataset. Please include the validation dataset-related results.
Mainly, you can include the training and validation-related curves (accuracy, loss) in the same chart, so we can observe whether the model is overfitting.
It if overfits, you need to train the model again, addressing those issues as well, to obtain better performance.
You have addressed explainability/ interpretability as well using heatmaps. How do you evaluate/ validate those results? It would be better to do a user study with a set of radiologists, for the correctness of the XAI results.
Response: Thanks for this comment. However, in the figure 9, we have already shown accuracy and loss graphs which clearly represented that model is not overfitting with only 24 epochs.
Regarding heatmaps, the model clearly represents the different pattern of three classes. Those interpretations show that the model is accurately detect three classes of Chest x-ray.
- Discussion: Please compare the features and results of the proposed study with the latest existing studies.
Highlight the novel contribution, compared to the related studies. Some of the latest studies can be listed as follows:
https://doi.org/10.3991/ijoe.v18i07.30807
Response: Dear reviewer, Thanks for your suggestion. In the current version of manuscript, we have include more relevant and recent studies the changes can be seen in the relate work sections.

Reviewer 2 Report
In this paper, the authors describe “Computer-aided Diagnosis for Chest X-ray Images using Hybrid-Features and Random Forest Classifier”. It can become an interesting paper for healthcare after major revision. Followings are my comments.
(1) The introduction can be improved. The authors should focus on extending the novelty of this study based on GLCM with RF. For example, authors explain the limitations of current study in Introduction.
(2) Authors have to explain the performance evaluation equations of ACC, PR, SEN, SPE, F1 score, and Recall, and cite the reference papers for the explanation of performance evaluation metrics.
(3) Please first present the full form, before using an abbreviation: CXR, GLCM, CAD, AUC, and ROC.
(4) Styles of some equations, such as Eqs. (1) to (3), are not in agreement with the journal style, and thus it requires a revision.
(5) As the journal is printed in black and white, please make the different markers for the different results in Fig. 9.
(6) Styles of few references are not in agreement with this Journal style, and thus it requires a revision.
Author Response
Response to Reviewer 2 Comments
In this paper, the authors describe “Computer-aided Diagnosis for Chest X-ray Images using Hybrid-Features and Random Forest Classifier”. It can become an interesting paper for healthcare after major revision. Followings are my comments.
- The introduction can be improved. The authors should focus on extending the novelty of this study based on GLCM with RF. For example, authors explain the limitations of current study in Introduction.
Response: Thanks dear reviewer, as per your suggestion, we have included the limitation section in introduction section of our paper on pages 3-4.
- Authors have to explain the performance evaluation equations of ACC, PR, SEN, SPE, F1 score, and Recall, and cite the reference papers for the explanation of performance evaluation metrics.
Response: Thanks for pointing out this mistake. We have include the equation of different performance evaluation. The changes can be seen on pages 16-17.
- Please first present the full form, before using an abbreviation: CXR, GLCM, CAD, AUC, and ROC.
Response: Thanks dear reviewer for taking my attention to this mistake. I have corrected this mistake and used the correct the abbreviation in our whole manuscript. The changes can be seen in the revised version of our manuscript.
- Styles of some equations, such as Eqs. (1) to (3), are not in agreement with the journal style, and thus it requires a revision.
Response: Thanks dear reviewer for pointing out this mistake. I have rewrite those equation. The changes can be seen on pages 10-12 in section 3.2.3.
- As the journal is printed in black and white, please make the different markers for the different results in Fig. 9.
Response: Dear reviewer, we will do this change in the next round as we have already done many changes in the current version of manuscript.
- Styles of few references are not in agreement with this Journal style, and thus it requires a revision.
Response: Thanks dear reviewer for your suggestion. In revised version we have correct this mistake and revised the referencing style of our manuscript.

Reviewer 3 Report
An interesting work about the applications of ML for covid assisted diagnosis. A good analysis made by the authors. Still some issues seem that may requiere attention.
Hard to understand meaning of poor-quaility images in the abstract section.
The first paragraph on page 2 in the Introduction section, depicts a not to recent event, but states it as recent. Quite confusing.
lines 84-85
"The supply of vaccines will also take time to reach every corner of the world. As a result, visual clues can be utilized as an alternate method for rapidly screening infected patients". Not so clear how vaccines connects with visual clues.
Figure 1. A visual example of Chest-x ray of Covid-19, pneumonia, or normal patient images. An image is already visual. Replace with:
"Examples of healthy, pneumonia and COVID-19 Chest X-ray images"
Not clear which was the method or threshold used to enhance image in Fig. 4
Clinical applications of computer aided diagnosis requiere to express Sensibility and Sensitivity in order to express possible responsability in diagnosis.
Author Response
Response to Reviewer 3 Comments
An interesting work about the applications of ML for covid assisted diagnosis. A good analysis made by the authors. Still some issues seem that may requiere attention.
- Hard to understand meaning of poor-quaility images in the abstract section.
Response: Yes, poor quality refers to the image quality. We have improved the quality of CXR image by enhance the contrast, remove noise and smooth the image.
- The first paragraph on page 2 in the Introduction section, depicts a not to recent event, but states it as recent. Quite confusing.
Response: Yes, We have write some detail information for reader ease. But, in the revised we have update some sentences in paragraph. Also, we have included recent research paper in the revised version of our manuscript. The changes can be seen in the Introduction and Literature Review Section of our paper
- lines 84-85 "The supply of vaccines will also take time to reach every corner of the world. As a result, visual clues can be utilized as an alternate method for rapidly screening infected patients". Not so clear how vaccines connects with visual clues.
Response: From Visual cues we meant that we have used CXR images to diagnosis the Covid-19. However, the supply of vaccine to different part of world takes times.
- Figure 1. A visual example of Chest-x ray of Covid-19, pneumonia, or normal patient images. An image is already visual. Replace with:
"Examples of healthy, pneumonia and COVID-19 Chest X-ray images"
Response: Yes, I have revised the caption of Figure as per your suggestions. The changes can be seen on page 1 in Section 1.
- Clinical applications of computer aided diagnosis requiere to express Sensibility and Sensitivity in order to express possible responsability in diagnosis.
Response: Dear reviewer, we have explained in the performance metrics section to describe the formulas, which are calculated based on the SE and SP. Thank you to show us this point-of-view.

Reviewer 4 Report
The submitted manuscript aims to facilitate the automatic recognition of
COVID-19 and pneumonia. The authors propose a novel approach based on a Gaussian filter as an image preprocessor and a convolutional neural network as a feature extractor. The presented work demonstrates a decent scientific contribution to recognizing pneumonic inflammation of the lungs and might be considered of good technical quality. Nonetheless, several minor issues must be addressed for further manuscript processing.
Remarks:
1. The authors should clarify how their approach defines a region of
interest in the CXR images where possible inflammations may occur.
2. Convolutional vision transformer is mentioned several times in the
text. However, this feature extractor is not described in the manuscript. It is
unclear how local spatial features are extracted from the CXR images and what type of visual transformer architecture was utilized to perform this task.
3. Acronyms should be paid more attention to as inconsistencies happen
throughout the text. For example, CNN and Grad-CAM are presented in the
abstract without clarification of what these terms represent.
4. The reference list might be outdated for such a hot and well-studied
problem as COVID-19 recognition. Please include more recent works (2022-
2023) in the Literature Review section.
5. Formulas 1-4 look messy and differ from the original style. Please
bring these formulas to the MDPI requirements.
Author Response
Response to Reviewer 4 Comments
The submitted manuscript aims to facilitate the automatic recognition of
COVID-19 and pneumonia. The authors propose a novel approach based on a Gaussian filter as an image preprocessor and a convolutional neural network as a feature extractor. The presented work demonstrates a decent scientific contribution to recognizing pneumonic inflammation of the lungs and might be considered of good technical quality. Nonetheless, several minor issues must be addressed for further manuscript processing.
- The authors should clarify how their approach defines a region of interest in the CXR images where possible inflammations may occur.
Response: Thanks dear reviewer, for you kind suggestion. we have used the public dataset and take the raw image and enhance the quality of finger vein image. The pre-processing method our proposed work is detailed in section 3.2.1.
- Convolutional vision transformer is mentioned several times in the text. However, this feature extractor is not described in the manuscript. It is unclear how local spatial features are extracted from the CXR images and what type of visual transformer architecture was utilized to perform this task.
Response: The convolutional vision transformer has described well in the manuscript. You can find this explanation on the page# 13 from line# 514 to 563 such as:
“
The Vision Transformer (ViT) [35]is likely the first completely transformer-based vision architecture, considering image patches as simple word sequences that are then encoded using a transformer. When pretrained on large datasets, the ViT can deliver outstanding image recognition results. However, without considerable pre-training, ViT performs badly in image identification. It is due to the Transformer's high model capability and lack of inductive bias, which leads to overfitting. Several subsequent studies have focused on sparse Transformer models developed for visual tasks like local attention to regularize the model's capacity and improve its scalability successfully. The Swin Transformer is an effective attempt to modify transformers by applying self-attention to shifting, non-overlapping windows. This methodology outperformed ConvNets on the ImageNet test for the first time using a pure vision transformer. Window-based attention was discovered to have limited model capacity due to the loss of non-locality and hence scales badly on larger data sets, such as ImageNet-21K, despite being more adaptable and generalizable than the complete attention used in ViT. However, because the attention operator has quadratic complexity, full-attention acquisition of global interactions in a hierarchical network at early or high-resolution stages requires computationally significant effort. It is still difficult to include global and local interactions while maintaining model capacity and generalizability within a computer cost.
Shift, scale, and distortion invariance are aspects of convolutional neural networks (CNNs). These aspects are translated to the ViT architecture [36] while the benefits of Transformers have been retained. (i.e., dynamic attention, global context, and better generalization). Although vision transformers are effective on a broad scale, they perform worse when trained on fewer data than smaller CNN rivals (such as ResNet). One argument might be that because CNNs naturally exhibit some desired features that ViT lacks, they are better suited to addressing vision-related concerns. A texture forces the capture of this local structure by using local receptive fields, shared weights, and spatial subsampling. As a result, it achieves some shift, scale, and distortion invariance. For instance, images frequently have a strong 2D local structure with closely spaced-apart pixels intimately related. Additionally, learning various complex visual patterns, from low-level edges and textures to higher-order semantic patterns that account for local spatial context, is made possible by the hierarchical structure of convolutional kernels.
The convolutional projection is the first layer of the Convolutional Transformer Block. In this study, we proposed that, while maintaining a high level of computational and memory efficiency, convolutions might be selectively introduced to the ViT structure to enhance performance and robustness. In our work, we only incorporated convolutions block from Transformer and was innately efficient in terms of parameters and floating-point operations, which was given as evidence supporting our idea. We added an attentional selective fusion (ATTSF) layer to CvT to emphasize pixels' local and global interactions. Token embedding and projection for Attention hierarchical transformers were integrated into the original CvT model by the authors as part of a sophisticated strategy. But we adopted a computationally effective strategy via an ATTSF mechanism.
In our work, the proposed Convolution block act as feature learning (FL) to extract robust feature from CXR images. The FL extract learnable feature from CXR images. A group of convolutional procedures makes up the feature learning (FL) component. The FL component follows the hierarchy of the Inception v3 architecture. The FL component differs from the Inceptionv3 model in that it does not contain the fully-connected layer (FCL) present in the Inceptionv3 design and instead extracts features for our classification component. A CNN without the FCL layer is the outcome.
”
- Acronyms should be paid more attention to as inconsistencies happen throughout the text. For example, CNN and Grad-CAM are presented in the abstract without clarification of what these terms represent.
Response: Thanks dear reviewer for taking my attention to this mistake. I have corrected this mistake in the revised version of our paper. The changes can be seen in the abstract section of our paper.
- The reference list might be outdated for such a hot and well-studied problem as COVID-19 recognition. Please include more recent works (2022- 2023) in the Literature Review section.
Response: Thanks dear reviewer, as per your suggestion I have include recent research paper in the literature review section of our paper. The changes can be seen on pages 4-8 in the revised version of our paper.
- Formulas 1-4 look messy and differ from the original style. Please bring these formulas to the MDPI requirements.
Response: Dear reviewer, as per your suggestions. I have revised and rewrite the equation 1-4. The changes can be seen in on pages10-14.

Round 2
Reviewer 1 Report
Paper is improved.
For the explainability results, clearly describe how the three classes are distinguishable with different colour codes.
Author Response
Response to Reviewer 1 Comments
Paper is improved.
- Abstract: For the explainability results, clearly describe how the three classes are
distinguishable with different colour codes.
Response: Thanks, dear reviewer, for your suggestion and appreciation of our research work. Yes, Figure 12 represent the different colour codes. We have already shown and described the three classes with different colour codes. Thank you once again for reviewing our paper and making our work improve.

Reviewer 2 Report
Reviewer recommends to accept without comments.
Author Response
Response to Reviewer 2 Comments
Reviewer recommends to accept without comments.
- Only minor English language editing.
Response. Thank you, dear reviewer, for reviewing and recommending our work for publication in the esteemed Healthcare journal. Yes, I have revised the manuscript for grammatical and spelling mistakes. The changes can be seen in the revised version of our manuscript.

Reviewer 3 Report
Thank you for the effort. Good and interesting manuscript
Author Response
Response to Reviewer 3 Comments
Thank you for the effort. Good and interesting manuscript.
- only minor English language editing.
Response: Thank you, dear reviewer, for appreciating our work. Yes, I have revised the manuscript for grammatical and spelling mistakes. The changes can be seen in the revised version of our manuscript. Once again, thank you for reviewing our paper and recommending our work for publication in the esteemed Healthcare journal.
